# Emergent digital bio-computation through spatial diffusion and engineered bacteria

Alex J. H. Fedorec [1,4] ✉, Neythen J. Treloar[1,4], Ke Yan Wen[1,4], Linda Dekker [1], Qing Hsuan Ong[1], Gabija Jurkeviciute[1], Enbo Lyu[1], Jack W. Rutter [1], Kathleen J. Y. Zhang[1], Luca Rosa[1], Alexey Zaikin[2,3] & Chris P. Barnes [1] ✉

Biological computing is a promising field with potential applications in bio-safety, environmental monitoring, and personalized medicine. Here we present work on the design of bacterial computers using spatial patterning to process information in the form of diffusible morphogen-like signals. We demonstrate, mathematically and experimentally, that single, modular, colonies can perform simple digital logic, and that complex functions can be built by combining multiple colonies, removing the need for further genetic engineering. We extend our experimental system to incorporate sender colonies as morphogen sources, demonstrating how one might integrate different biochemical inputs. Our approach will open up ways to perform biological computation, with applications in bioengineering, biomaterials and biosensing. Ultimately, these computational bacterial communities will help us explore information processing in natural biological systems.

Engineering biological systems capable of computation has long been a goal of biology[1]. The first synthetic biology papers engineered a toggle switch[2], oscillator[3], and autoregulation[4], which can be used as fundamental components in engineering a computer[5]: memory, clock, and noise filter. The most common paradigm in biological computation to date has been the engineering of programmable logic into gene regulatory networks (GRNs) inside cells[1]. The current state-of-the-art are automated design tools such as Cello[6–8], which compiles a digital function into a DNA sequence enabling a comprehensive suite of three input digital functions to be constructed.

The complexity of genetic circuits that can be programmed into a single cell is limited by the number of available orthogonal components and metabolic burden[9,10]. In addition, each new function requires extensive genetic engineering. To overcome these limitations, computational communities of microbes have been developed in liquid culture[11–13]. This allows the division of a complex function into smaller, isolated modules that can be programmed into different populations and integrated using inter-population communication[14]. However, building microbial communities that communicate in liquid culture requires the use of a unique communication molecule for each "wire" and the avoidance of crosstalk between molecules[15].

The limitations of monoculture computing and distributed communities in liquid culture can be overcome by producing logic gates based on spatially arranged, communicating bacterial colonies in solid culture. The GRNs inside each colony are simple and isolated, and the signal produced by a colony is spatially localized. This allows for the construction of complex digital functions without the wiring problem of liquid culture communities. Early work used four different quorum molecules to connect bacterial NOR gates[16], and another work decomposed digital functions into the sum of spatially separated modules[17]. Subsequent research used the spatial arrangement of sender cells and modulator cells to print digital functions on a piece of paper[18]. That work enabled the printing of digital functions using stamps and "cellular ink", and demonstrated that the printed functions were storable for later use[19].

In this work, we combine simple inducer activation functions with spatial structure to produce modular, easily programmable bacterial computers. In contrast to previous work[16–18], much of the information processing is encoded within interacting morphogen gradients,

[1]Department of Cell and Developmental Biology, University College London, London WC1E 6BT, UK. [2]Department of Mathematics, University College London, London WC1E 6BT, UK. [3]Institute for Women's Health, University College London, London WC1E 6BT, UK. [4]These authors contributed equally: Alex J. H. Fedorec, Neythen J. Treloar, Ke Yan Wen. ✉e-mail: a.fedorec@ucl.ac.uk; christopher.barnes@ucl.ac.uk

enabling us to program arbitrary digital functions with minimal genetic engineering. Our computational devices consist of bacterial receiver cells positioned relative to sources of diffusible molecules—either a droplet of inducer or bacterial sender cells. Each source represents an input to the computation, and the receivers produce a fluorescent output that represents the result of the computation. We demonstrate the position dependence of our activation functions using lawns of receiver cells and droplets of IPTG as diffusible inputs. Then we demonstrate the construction of two-input digital functions using colonies of receiver cells. We show that each receiver can be programmed to encode many different two-input logic gates by changing their position relative to the inputs. We present a mathematical framework to understand these results and make mathematical statements of its capability. Using this framework, we developed an algorithm that minimizes the complexity of the spatial functions and greatly reduces the required genetic engineering. This approach also guarantees that any function can be implemented using only one communication molecule, solving the wiring problem. Finally, we demonstrate the use of biosensor colonies as sources of the diffusible molecule, enabling computation based on multiple environmental inputs and providing a simple proof-of-concept for future applications such as medical diagnostics or pollution testing.

## Results

### Analog-to-digital conversion of interacting diffusion patterns allows for spatial computing

Inputs and outputs of a digital function can be ON (1) or OFF (0). For a two-input digital function this results in four possible input states—00, 01, 10, 11—each of which can produce a 0 or 1 output. There are 16 unique ways of mapping the four input states to the binary output, resulting in a total of sixteen unique two-input logic gates. In our computation paradigm, inputs are sources of diffusible molecules that produce a high concentration near to the source that decays as one moves further away, with 0 and 1 indicating the absence or presence of each source, respectively. All of our inputs produce the same diffusible molecule, so where there is overlap in the diffusion field, the resulting concentration is additive. These inputs produce a unique diffusion field for each of the four input states. Bacterial "receiver" cells, capable of sensing and responding to the concentration of the diffusion field at their specific locations, map the input states to binary outputs.

Here, we consider four ways in which receiver cells can respond to increasing concentrations of the diffusible molecule: a "highpass", where the cells are switched on above a certain concentration; a "bandpass", where cells are only on at intermediate concentrations; and their inverses, a "lowpass" and "bandstop". In a homogenous field of receiver cells, the diffusion field produced by the two inputs will trigger a response in each of the cells depending on where they are located, resulting in a specific response pattern. Using a toy model simulated with the finite difference method[20], we demonstrate that a diffusion field, interpreted using our set of activation functions, produces digital logic (Fig. 1A, B). For each of the input configurations, a simulation was performed in which each input is a source of diffusible molecule added at the beginning of the simulation. The four activation functions were applied to the resultant diffusion fields, and the digital functions produced at each position were determined. This results in a map where the color of each region represents the logic gate that would be encoded if a receiver cell were placed in that region. These simulations show that with the bandpass and bandstop alone, it is possible to encode all sixteen two-input digital logic gates. It is also clear that the logic gate encoded by a given receiver is dependent on its position relative to the inputs.

For a simple receiver, in which the cells turn on when the concentration is above a threshold, when the two input sources are close to each other, within the overlapping region, the concentration will be high enough to trigger the output when one or both inputs are present

(Fig. 1E). If the two input sources are moved further apart, at the center of the overlapping region the concentration will not be high enough to trigger the output when only one source is present, but when both sources are present, the sum of the concentration will push the concentration above the output threshold. As such, we are able to change the logic function computed, by moving the sources of the diffusible molecules.

### Engineered *E. coli* can generate predicted digital logic from diffusion patterns

To implement the system experimentally, we utilized genetic circuits that respond to IPTG to form either highpass or bandpass receiver responses through the expression of GFP[21]. The bandpass behavior is produced by a T7 promoter that is repressed by the PhlF transcription factor (Fig. S3). T7 RNAP and PhlF are expressed from IPTG inducible promoters in the genome and on a plasmid, respectively. In the absence of IPTG, no T7 RNAP is expressed, and the output promoter is not activated. At intermediate levels of IPTG, T7 RNAP is produced, and transcription from the output promoter occurs. At high IPTG levels, PhlF is expressed at a high enough concentration to inhibit transcription from the output promoter. The highpass receiver is the same system but without the PhlF-expressing plasmid. Their dose responses were characterized in liquid culture (Fig. S3).

We sought to understand how the IPTG diffuses and how the bacteria growing on an agar substrate respond. Agar plates were seeded with our engineered cells, and we dispensed droplets of IPTG at different concentrations (Fig. S4A). The diffusion field created by the IPTG droplet depends on its concentration and the time allowed for it to diffuse. One might, therefore, expect the diameter of the ring formed by the bandpass cells to change with concentration and time. Indeed, with higher inducer droplet concentrations, the ring size increases. However, the observable position of the ring is temporally static, indicating that at some point—perhaps upon reaching a particular growth phase—the cellular response is established and fixed.

With an understanding of the diffusion field and the response of our cells, we can design spatial patterns of inputs that can produce specific logic outputs. Indeed, with this information, we can place droplets of IPTG inducer at defined positions on a field of cells and demonstrate that all the predicted regions present for the highpass and bandpass receiver can be observed in practice (Fig. 1C, D). Simply by changing the distance between the two inputs, we change the position of specific logical responses to those inputs (Figs. 1F and S4B).

### Precisely positioned engineered bacterial colonies serve as digital function outputs

Given that we can observe different digital functions at specific positions within a field of cells, we hypothesized that it should also be possible to place colonies at defined locations and interpret their output response as a digital logic gate. The use of colonies instead of a field of cells simplifies the read-out, as the position from which to read the output has been pre-defined by the colony placement. Further, it enables the use of different receiver cell types at different positions, which will allow us to construct more complex logic gates.

We used a liquid handling robot to dispense droplets of culture that form into colonies (Fig. 2A). This restricts us to positions defined by a 384-well microtiter plate layout. We characterized the responses of bandpass and highpass colonies at defined distances from droplets of different concentrations of IPTG input (Fig. S3). With this data, we were able to fit a reaction-diffusion model that incorporates the growth of the bacterial colonies, diffusion of the IPTG inputs, and the dynamics of the transcriptional networks within the cells (Supplementary Information 4 and Fig. S12).

We designed spatial patterns of inputs and receiver colonies that are capable of producing the four, non-trivial, two-input logic gates achievable with the highpass and bandpass (Fig. 2B). We then

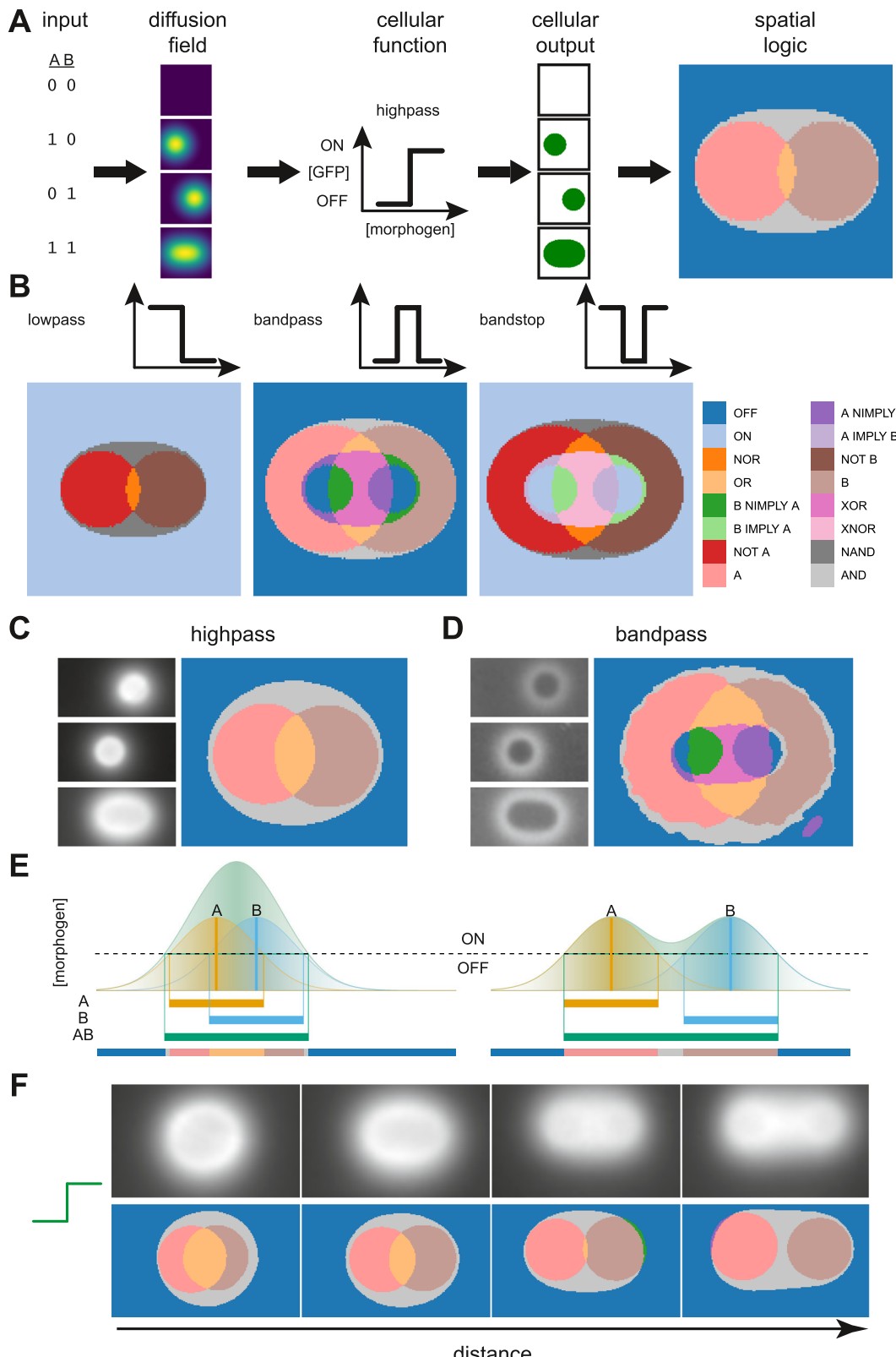

implemented these spatial patterns experimentally in 6-well plates using droplets of IPTG for inputs and colonies of bacterial receivers as outputs (Fig. 2C). The growth and fluorescence of each colony was determined using a custom imaging system (Methods). The fold change in fluorescence was calculated, and each gate was scored by dividing the lowest ON state by the highest OFF state[6]. All of the gates perform their predicted function, though there are deviations from the

model-predicted fluorescence levels. One can clearly see the high fluorescence output of the highpass colonies compared with the bandpass colonies, leading to a very high score for the highpass OR gate. The AND gate does not perform as well, and the reason for this can be intuited from the dose-response curve of the highpass (Fig. S3). A doubling of inducer concentration—as when moving from one input being present to both inputs present—cannot result in a shift from

**Fig. 1 | Logic gates can be produced from a combination of spatial position and engineered response function. A** Given two possible digital inputs, four combinations can be produced. If each input corresponds to a diffusible molecule, each input configuration produces a unique gradient pattern. Cells are engineered to respond to the concentration of the diffusible molecule in a specific way—in this example, turning on above a threshold. Cells at each location in the environment turn on or off depending on the input configuration and their engineered function. Each location can then be mapped to a specific logic gate, given the cells' responses. **B** Different response functions will produce different logic gates. All 16 two-input logic gates can be produced with bandpass and bandstop response functions.

**C, D** Response of a lawn of highpass and bandpass receiver cells induced with droplets of IPTG at either **A**, **B**, or **A** and **B** input locations. Images are of plates after 20 h. The images are processed, and a digital function is assigned to each pixel. **E** The gradients produced by two sources of the same diffusible molecule, located at **A** (orange) and **B** (blue), are additive where they overlap (green). The regions above a threshold are considered to be ON. If the distance between the sources is changed, the location of the ON regions changes correspondingly, producing different logic. **F** Response of a lawn of highpass receiver cells induced with droplets of IPTG and their corresponding digital functions as the distance between the IPTG locations increases.

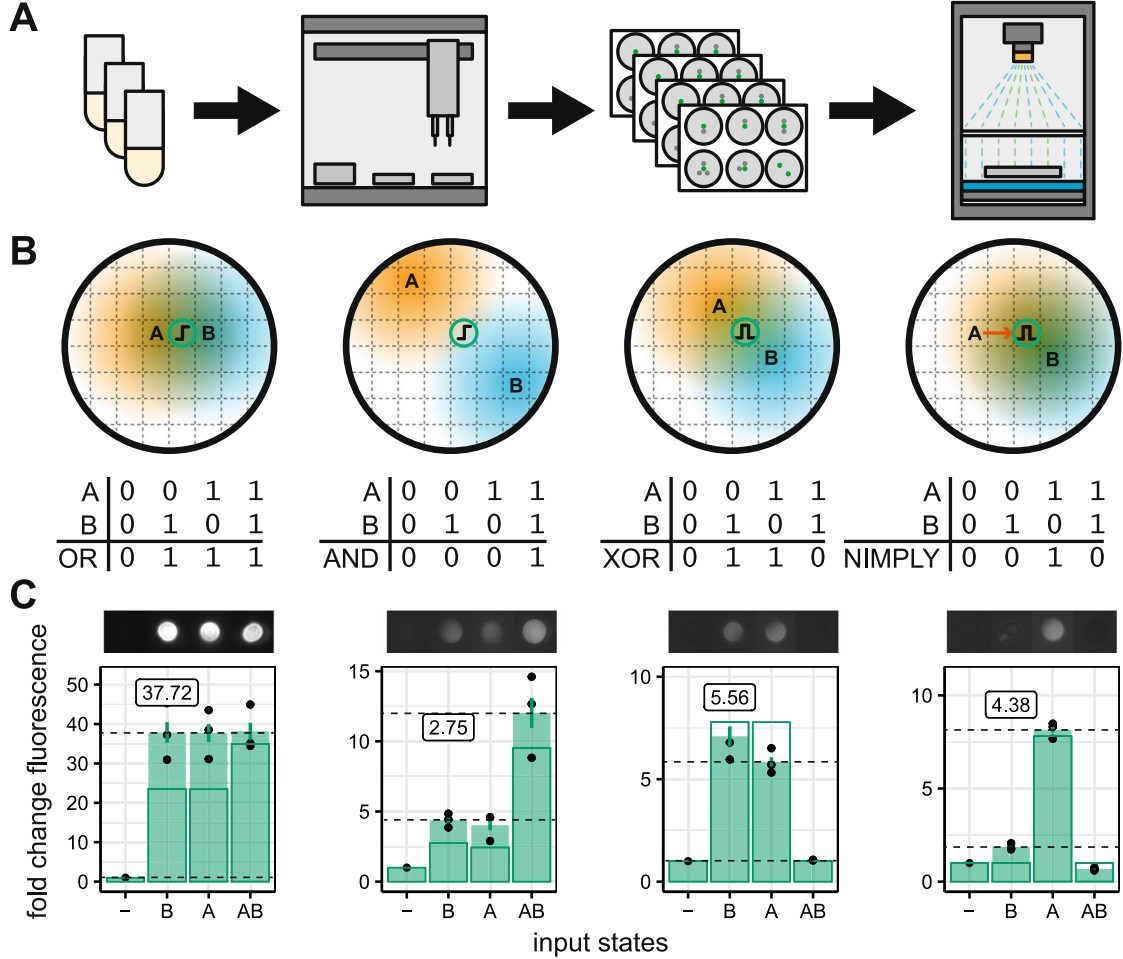

**Fig. 2 | Biological implementation of the spatial computing framework.**
**A** Receiver colonies and IPTG inputs were dispensed at pre-defined positions using a liquid handling robot and imaged for growth and fluorescence. **B** Predicted spatial layouts that will give rise to OR, AND, XOR, and NIMPLY logic responses. Letters indicate the position of the respective IPTG inputs, and circles containing highpass or bandpass cartoons indicate the location of output colonies. Colored regions represent the diffusion of the IPTG from the input locations. **C** Colony responses to all input configurations at 20 h. Images show a representative colony for each configuration (highpass OR exposure = 40,000, intensity = 3, all other gates exposure = 60,000, intensity = 5). Solid bars show the mean fold change in fluorescence relative to a colony with no IPTG inputs. Error bars show the standard error of the mean for three replicates. Outlined bars show model predictions. The gate score displayed on the plot shows the least fluorescent ON state divided by the most fluorescent OFF state (shown with dashed horizontal lines).

minimal expression to maximal expression due to the shallowness of the dose–response curve. The best we can achieve with our highpass is approximately threefold fluorescence change.

**An abstract representation of receiver response to signaling gradients facilitates an analysis of realizable digital functions**
Up to this point, the logic gates that we have implemented have been simple enough that sensible positions for inputs and outputs can be inferred from the receiver characterization data. However, as we wish to construct more complex logical functions, a computational approach is required. To this end, we developed an abstract

representation of the inducer concentration seen by a receiver for each input state, allowing us to investigate the capabilities of the system (Supplementary Information 1.1).

We simplify the description of the spatial arrangement of the receiver and sources by considering a single dimension; the signal concentration observed by the receiver. The presence of a source of inducer increases the signal concentration seen by the receiver by an amount proportional to the distance between the receiver and source. By moving the receiver closer to a source, we increase the concentration seen by the receiver. To reflect this change, the input states that include that source move further up our signal concentration

dimension. Similarly, moving away from a source does the inverse. Figure 3A shows that moving a receiver relative to two sources changes the order of the input states on the signal concentration dimension.

We represent the activation functions—highpass, lowpass, bandpass, and bandstop—as boundaries partitioning these ordered input states into ON and OFF boxes (Supplementary Information 1.2). Figure 3A (and Fig. S1) shows how this partitioning of input states results in logic output. By moving the receiver and sources, we move the input states in and out of the ON and OFF boxes. In this way, we can change the logic gate encoded by different patterns of inputs and receivers. Using this representation, we can enumerate all the distinct logic gates that are possible, given a number of inputs and the available receivers, while obeying physical constraints. All two-input digital functions can be computed spatially using a single receiver, provided we have receivers with response functions given in Fig. 1A, B (Supplementary Information 1.3), corroborating the simulation results.

### Two-level logic enables the construction of arbitrary digital functions

We extended our theoretical analysis of spatial computing to three-input logic gates and found that we can only encode 152 of the 256 possible three-input logic gates with a single receiver output (Table S2). This demonstrates a single receiver output is not sufficient to enable the construction of arbitrarily complex digital functions. To overcome this limitation, we developed an algorithm for the optimization of two-level spatial digital logic. In electronic design, logic minimization is used to simplify a digital circuit into a minimal canonical form. The Espresso algorithm is a heuristic optimizer that attempts to minimize a function into the minimal set of OR of ANDs, where each AND term can contain any of the inputs or their negations[22] (Fig. 3B, top).

Taking inspiration from the Espresso algorithm, we have developed an algorithm named the Macchiato algorithm, which finds the minimal set of receivers that encode a given digital function. The Macchiato algorithm takes the truth table of a multi-input, one-output function and returns the approximate minimal set of bandpass, highpass, lowpass, and bandstop receivers that are required to construct the digital function and the required mapping of input states for each receiver. The minimal set of receivers constitutes the analog of the AND layer and the mapping represents the connectivity to the inputs. The OR can then be performed using a highpass receiver that activates if any of the previous layers is activated (Fig. 3B, middle), or it can be done implicitly in the sense that if any individual receiver is ON, we take the output as ON (Fig. 3B, bottom). It is for this reason that we focus on an analog of the OR of ANDs (sum of products) form of a digital function, as we gain the flexibility to encode two-level digital functions using only one level of biological signaling.

For each three-input gate, the number of bacterial colonies required to build the gate was calculated for different sets of available activation functions (Table S4). With all the activation functions, all three input logic gates are possible and only require one or two colonies. If we remove the bandstop from the available activation functions we retain the capability to do all the three-input logic gates, however a larger proportion now require two bacterial colonies. If we also remove the bandpass activation we lose the capability to compute the majority of the three input logic gates.

The final step is to determine the optimal spatial positions of the inputs and receivers relative to one another. To do this, we use a calibrated finite difference model and an evolutionary algorithm, optimizing the difference in expected fluorescence between ON and OFF states (Fig. 3C).

### Designing complex spatial logic gates using the Macchiato algorithm

We demonstrate our approach by constructing a selection of three input logic gates (Fig. 4). These are designed using the Macchiato

algorithm to find the minimal set of highpass and bandpass receivers that will encode the given function. We started with a set that was capable of being realized with a single receiver colony and that covered a range of complexity if one were to implement them using state-of-the-art gene regulatory network designs[6]: $0 \times 7F = 27$ genetic parts, $0 \times 37 = 32$, Majority = 49. We also included two that had not been implemented previously: sum = 1 and the 4-input (A OR C OR D) AND NOT B. All of the logic gates function as expected. As with the two-input gates demonstrated above, the logic gates that require AND functionality from the highpass receivers—for example, $0 \times 37$ requires A AND C behavior—score lower than the other logic gates.

We then selected four logic gates that require two output colonies, as it is not possible to minimize the output from the truth table to a single block (Fig. 5). Two of these have been implemented previously using gene regulatory networks: Multiplexer = 38 parts, Rule 30 = 44. Rule 110 is a cellular automata rule set that is Turing complete. All the logic functions were successfully demonstrated. For example, in Fig. 5A, a bandpass receiver produces A > C ($0 \times 0A$) and a highpass receiver produces B AND C ($0 \times 11$); the OR of these two outputs produces IF C THEN B ELSE A, otherwise known as a 2-to-1 multiplexer with C as the selector ($0 \times 1B$). The digital function, again, performs as expected, however, the scores for both receiver colony functions are relatively low. The highpass is performing an AND operation which does not score highly, as previously stated. In addition, the A input position relative to the bandpass receiver is not in its ideal position, resulting in slightly lower activation. This positioning was probably produced by our algorithm to increase the distance of A from the highpass receiver, to prevent unwanted activation. Similar considerations likely result in the lower performance of the B XOR C operation in the $0 \times 6F$ gate (Fig. 5B).

### Using sender colonies as inputs for spatial digital logic

Up to this point, in order to demonstrate the computational approach, we have used robotically placed droplets of diffusible inducer as our function inputs. This is a rather restricted use case, limiting us to computations with a single input molecule and requiring printing of the computational device immediately prior to performing the computation. To remove these limitations, we move to a paradigm in which biosensing sender colonies act as sources of the diffusible molecule; producing it when triggered by the presence of an environmental input.

We constructed two *E. coli* strains that produce the diffusible quorum sensing molecule 3OC6-HSL, in response to chemical stimuli: arabinose and lactate. In addition, we built new receiver *E. coli* strains that respond to 3OC6-HSL with highpass and lowpass activation functions. The senders and receivers were characterized by solid culture (Figs. S6 and S7). A lowpass and highpass receiver were selected that responded at similar inducer concentrations.

From the characterization data, we determined patterns of sender and receiver colonies that produce the four non-trivial two-input logic gates that are achievable with highpass and lowpass outputs (Fig. 6). Two of these—NAND and IMPLY—require two output colonies, which are computed from combinations of NOT and IDENTITY functions using the OR paradigm described previously (Fig. 6D, E). A lower maximal fold-change for the highpass receiver, limits the score that can be achieved for gates using this strain as an output, however, all of these gates are successfully demonstrated.

### Discussion

We have shown that interacting spatial morphogen gradients, combined with analog-to-digital conversion, can encode complex biological computations. In our system, digital logic functions were built by patterning bacterial colonies, engineered to respond to diffusible molecules. We have proven that a single computing colony is capable of being programmed with any two-input logic gate by altering its

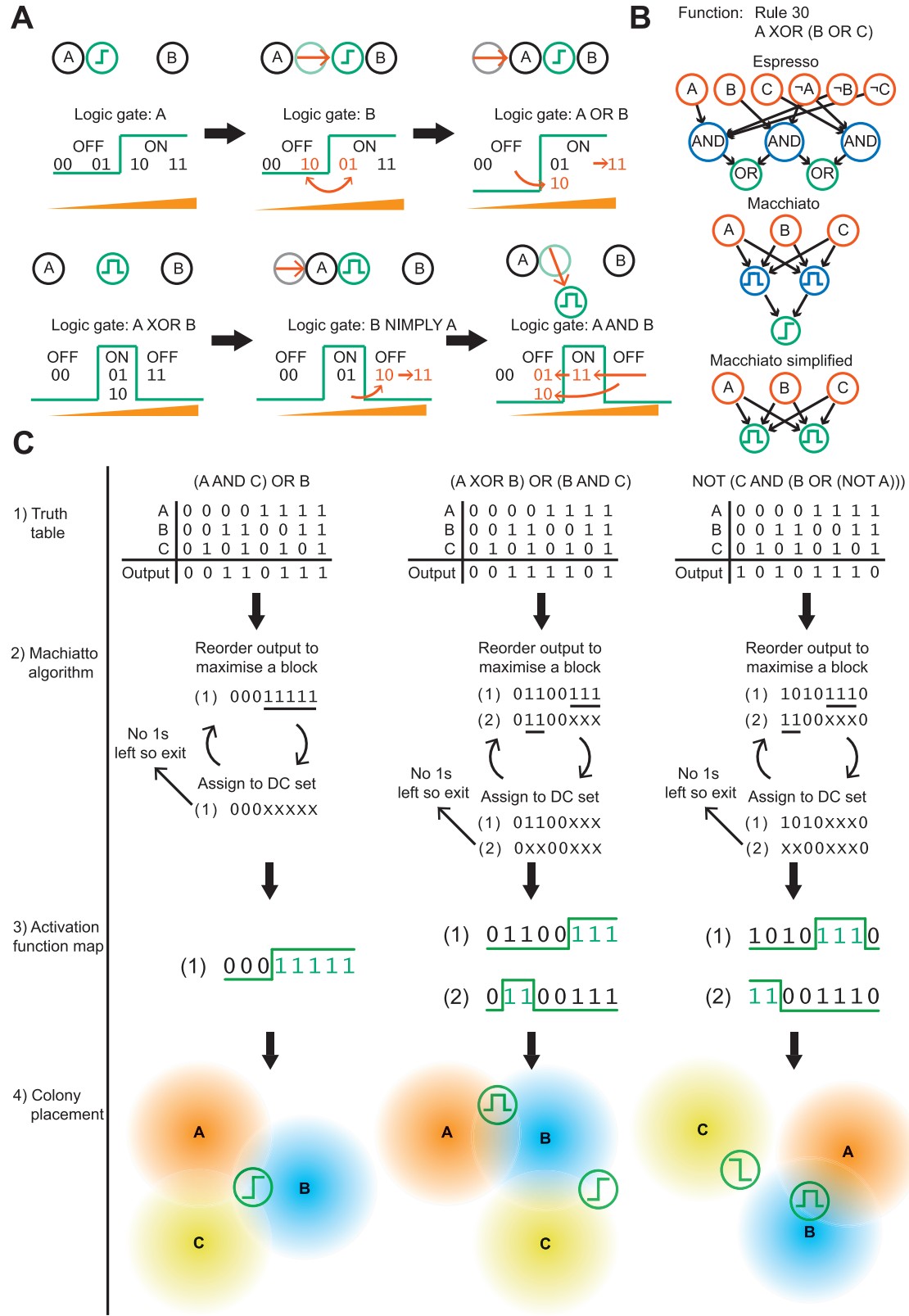

**Fig. 3 | Algorithmic determination of input and output patterning.**
**A** Demonstration of how varying positions and activation functions result in programmable digital logic and how this is represented in our framework. **B** A digital logic function simplified by the Espresso algorithm (top), Macchiato algorithm with two levels of biological signaling (middle) and Macchiato with one level of biological signaling (bottom). **C** The stages of the Macchiato algorithm on three examples, from left to right; (1) a truth table is supplied as input, (2) the input states are rearranged to minimize the number of blocks while obeying the constraints imposed by the system (DC = "don't care"), (3) activation functions are mapped onto the blocks, (4) spatial configuration of nodes is produced that results in the truth table, if any of the green output colonies are activated the output of the function is ON.

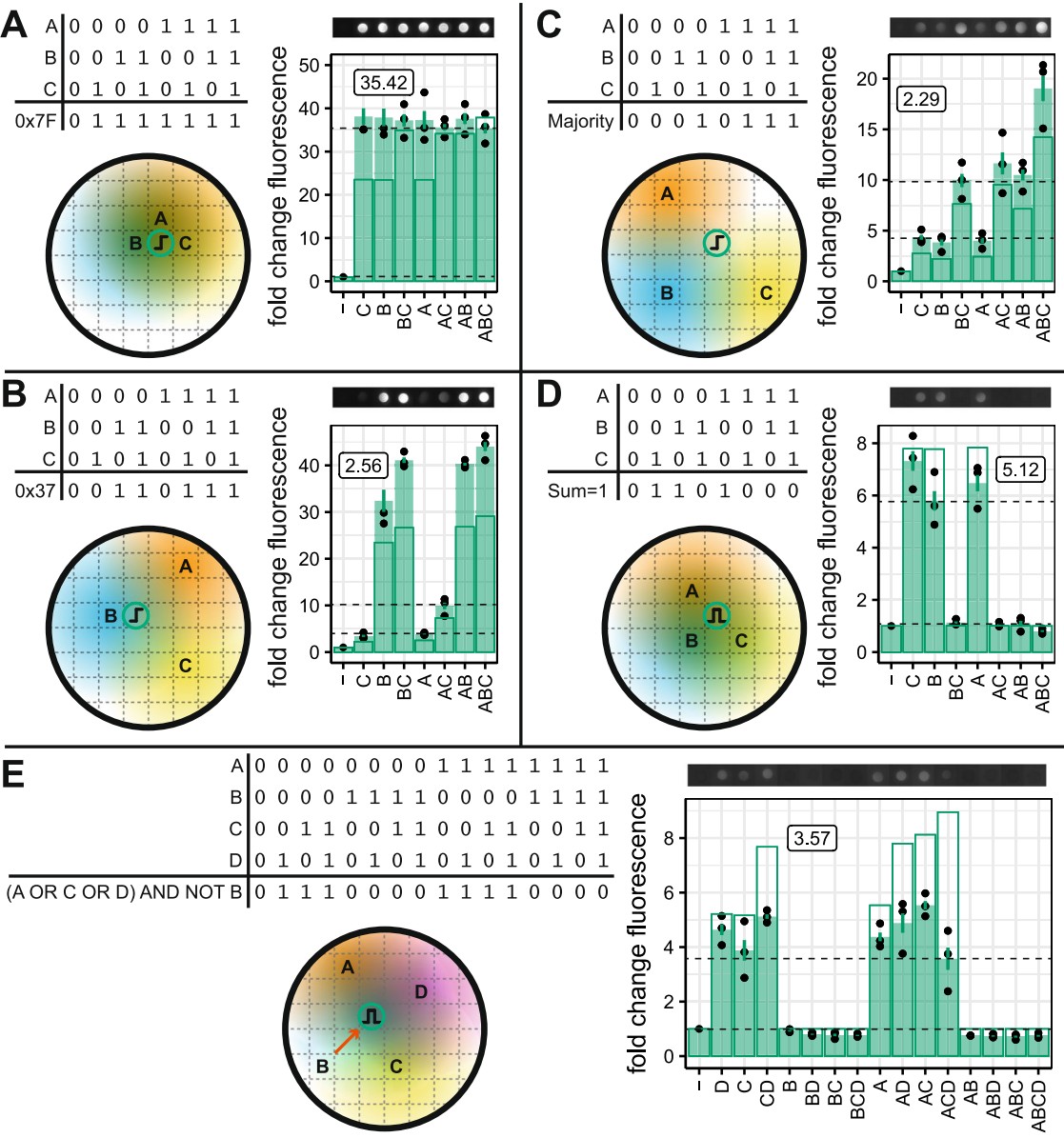

**Fig. 4 | Computing three-input and four-input spatial logic.** A selection of **A**–**D** three-input and **E** four-input logic gates with their output. Letters indicate the location of IPTG inputs, and the shaded regions represent the diffusion of the IPTG. Circles containing highpass or bandpass cartoons indicate the location of output colonies. Colony images show representative fluorescence intensity at 20 h (0 × 7F and 0 × 37 exposure = 40,000, intensity = 3, for all others exposure = 60,000, intensity = 5). Solid bars show the mean fold change in fluorescence relative to a colony with no IPTG inputs. Error bars show the standard error of the mean for three replicates. Outline bars show model predictions. The gate score displayed on the plot shows the lowest fluorescent ON state divided by the highest fluorescent OFF state (shown with dashed horizontal lines).

position relative to its inputs or replacing it with an alternative activation function. This was demonstrated using both lawns of bacteria and precisely positioned bacterial colonies. The transition from lawns to colonies is important as it simplifies the measurement of the output; it is easy to measure the fluorescence of a colony, but knowing which region of a lawn encodes a function is non-trivial. Further, the use of colonies allows us to use bacteria with different activation functions in different positions, enabling the construction of multi-output and multi-layered computations.

Additional complexity is necessary for the implementation of all three-input logic gates. We proposed an extension to our method, inspired by two-level electronic digital logic optimization. This required us to develop an algorithm for an analogous spatial digital optimization procedure to find the minimal set of bacterial colonies

required for a given function. We extensively analyzed this technique to find the requirements for the activation functions and provide a breakdown of the computational capabilities of the approach. We constructed a number of three-input logic gates, and demonstrated our approach is simpler than a state-of-the-art method to produce these functions in a gene regulatory network[6].

Incorporating sender colonies, capable of producing the diffusible molecule that the receivers respond to, expands the applicability of our computing system. Using biosensing senders that are triggered by environmental inputs may enable the development of computational devices that can be used for medical or environmental diagnostics. The use of senders removes another limitation; directly printing the diffusible molecule inputs, as we did in Figs. 1, 2, and 4 with IPTG start the computation immediately as the molecule begins to

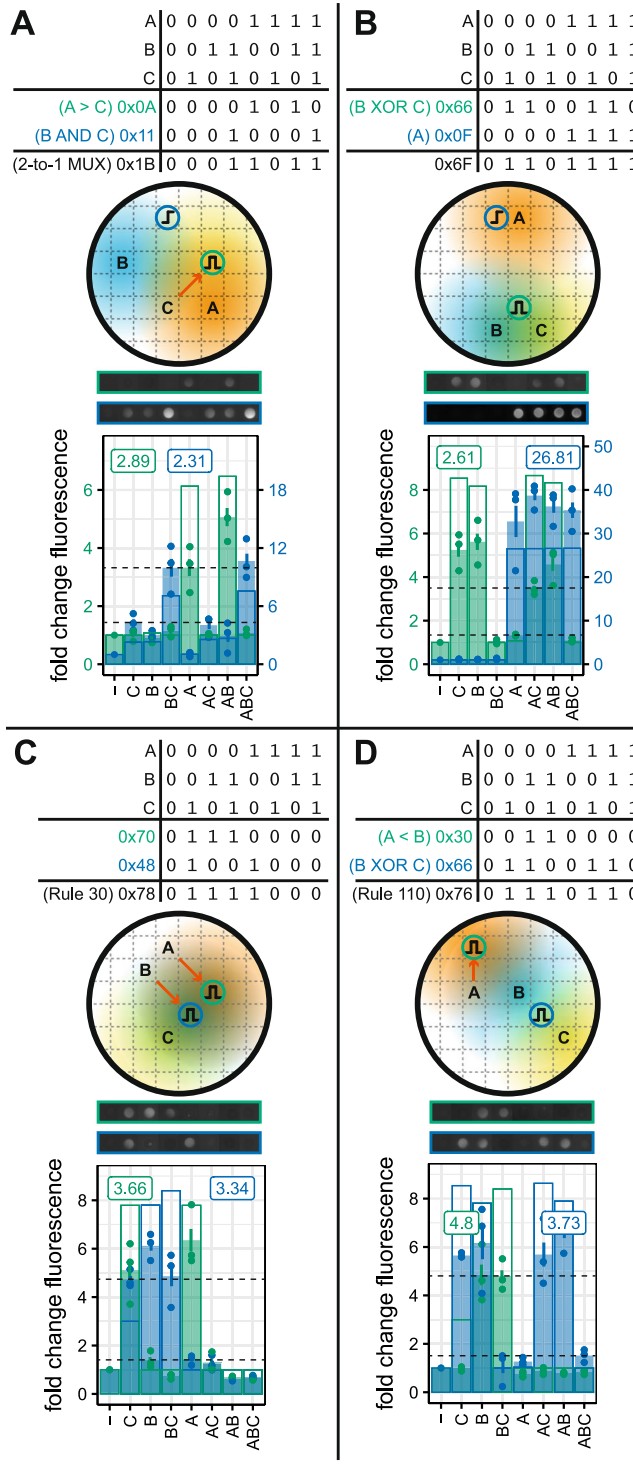

**Fig. 5 | Multiple output colonies enable complex logic. A–D** A selection of three input logic gates requiring two output colonies. The computation output is the implicit OR of the two output colonies i.e., if either of the outputs is ON, the computation is considered to be ON. Letters indicate the location of IPTG inputs, and the shaded regions represent the diffusion of the IPTG. Circles containing highpass or bandpass cartoons indicate the location of output colonies. Colony images show representative fluorescence intensity at 20 h (0 × 0F exposure = 40,000, intensity = 3, all others exposure = 60,000, intensity = 5). Solid bars show the mean fold change in fluorescence relative to a colony with no IPTG inputs. Error bars show the standard error of the mean for three replicates. Outline bars show model predictions. The gate score displayed on the plot shows the lowest fluorescent ON state divided by the highest fluorescent OFF state (shown with dashed horizontal lines).

diffuse. By printing with sender colonies at the locations of the input sources, it may be possible to pre-prepare devices in which the computation will only begin once the environmental stimuli are added. There has been great interest over the last decade in pre-prepared, bio-engineered, diagnostic devices[23].

In this work, we have taken advantage of genetic circuits that have already been engineered to produce two of our required activation functions[24]. However, in order to generate all possible digital functions, we would require a bandstop or lowpass genetic circuit. Our theory shows that we need, at most, $n-1$ output colonies to produce all $n$-input logic functions. This scales favorably compared to other biological computation approaches. However, as we add more inputs and outputs, it is feasible that it is not possible to find a valid pattern in two dimensions, as moving one input relative to one output also moves it relative to all other outputs. This may be alleviated by the redundancy present in the four activation functions i.e., it is possible to produce an OR gate in multiple ways using highpass or bandpass activation. Moving to three-dimensional devices would relieve this limitation, and provide an avenue for the development of computational bio-materials[25,26]. An additional challenge when we scale the number of inputs is that we amplify our weakness in computing AND logic. For two inputs, our receivers need to flip from the OFF to ON state with a doubling of signal concentration e.g., the difference between having one input present (01) and both inputs present (11). However, for three inputs, the same behavior needs to occur with only a 50% increase in signal concentration (011 to 111), and for four inputs, there is only a 25% increase (0111, 1111). This limitation in AND performance results in lower performance of some of the three and four input circuits, such as the 0 × 37 and majority circuits (Fig. 4B, C). Despite this limitation, we have successfully shown that the architecture of spatial computing is sound. In order to ensure that we can achieve a better implementation, the activation functions of our receiver colonies need to be as step-like as possible. Approaches such as the addition of positive feedback[27,28], toggle switch topologies[19] and recombinases[29] have been demonstrated in the past to improve digital response. This would provide cleaner transitions between ON and OFF regions, improving the signal-to-noise of our additive functions.

In comparison with previous, related approaches[16–18] our system allows the construction of any Boolean function with less biological complexity. This is because our choice of activation functions drastically offloads complexity from the internal cell dynamics to the morphogen field. Another advantage is the modular design that requires no genetic engineering to create a new biological computer once the initial set of activation functions has been produced. This also opens the door to the use of different cell types and inter-kingdom computational devices. These properties make our platform an appealing approach for biological computing and potential real-world applications. This work also emphasizes how the physical properties of systems can encode information, a fact exploited by early analog computers. Although spatial aspects have been explored in computer science, for example amorphous computing[30], utilization of space for information processing in biological systems could be a more general principle.

## Methods

### Strains and plasmids

Strains and plasmids used in this work are listed in Supplementary Table 5. We produced a number of strains capable of producing 3OC6-HSL in response to a molecule inducer or responding to 3OC6-HSL with GFP expression. These strains were produced using golden gate assembly and parts from the CIDAR MoClo[31], from the CIDAR MoClo extension (Richard Murray, Addgene), or the BioBrick library. Overnight cultures were grown in 15 mL Falcon tubes with 3 mL M9 glycerol media [1× M9 salts (BD Difco 248510), 2 mM MgSO$_4$, 0.1 mM CaCl$_2$, 0.4% glycerol, 0.2% casamino acids (MP Biomedicals 113060012),

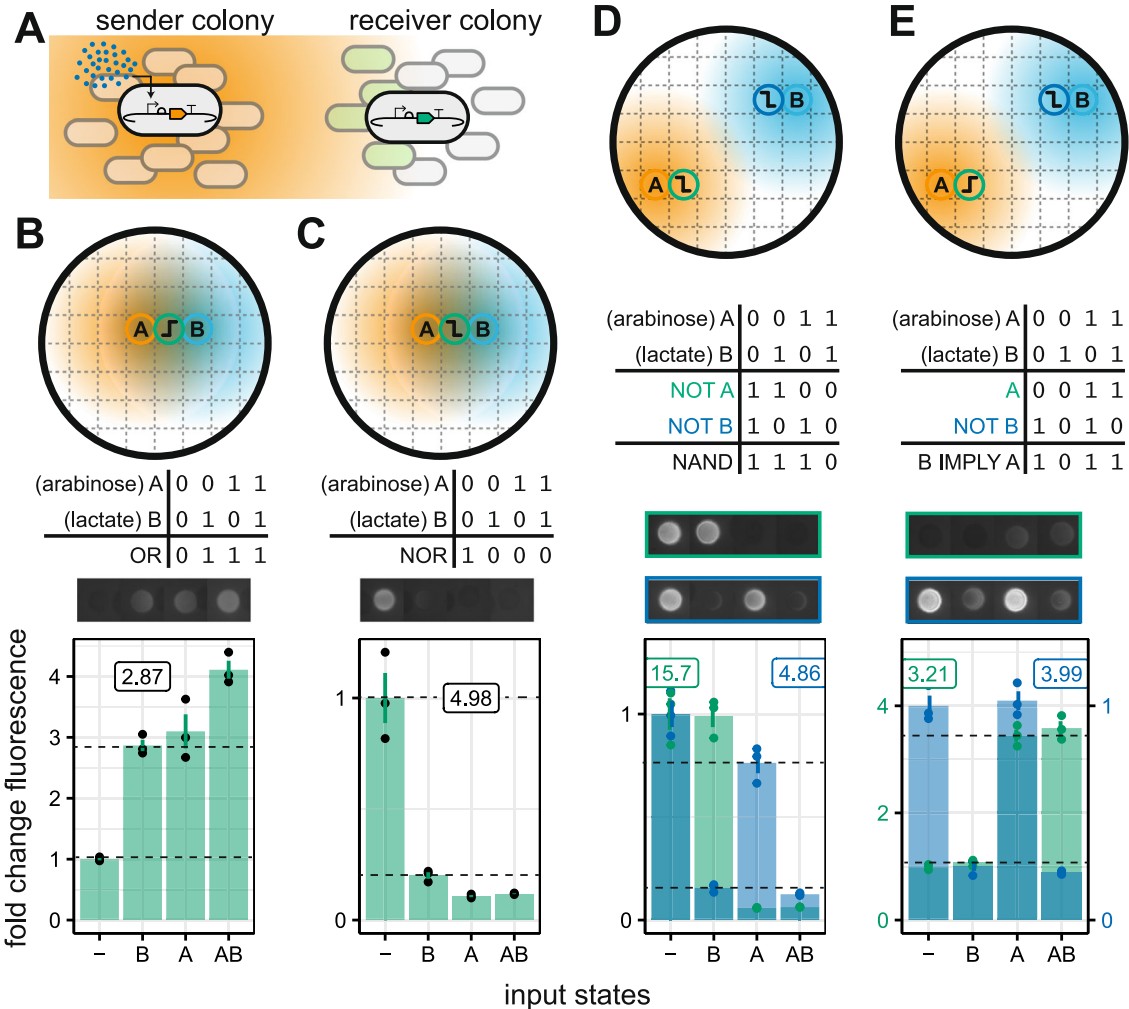

**Fig. 6 | Sender colonies as morphogen sources for spatial computation.**
**A** Biosensors, engineered to produce 3OC6-HSL in response to an environmental stimulus, were used as digital inputs. **B**–**E** Predicted spatial layouts that will give rise to OR, NOR, NAND and IMPLY logic responses. Circles containing letters indicate the position of the biosensor input colonies, and circles containing highpass or lowpass cartoons indicate the location of output colonies. Colored regions represent the diffusion of 3OC6-HSL from the input colonies. Colony images show

representative fluorescence intensity at 20 h (all gates exposure = 60,000, intensity = 5). Solid bars show the mean fold change in fluorescence relative to a colony with no inducer inputs to the senders. Error bars show the standard error of the mean for three replicates. The gate score displayed on the plot shows the least fluorescent ON state divided by the most fluorescent OFF state (shown with dashed horizontal lines).

0.035 mg mL-1 thiamine] with the relevant antibiotics at 37 C and shaking at 200 rpm.

### Liquid culture characterization

The optical density of overnight cultures was measured at 600 nm, and the cultures were diluted to an OD600 of 0.05. One hundred and twenty microlitres of the diluted culture was pipetted into a 96-well microtitre plate (Greiner 655096). The plate was covered with a plastic lid and incubated in the plate reader (Tecan Spark) for 2 h at 37 °C and shaking. The plate was removed from the plate reader after 2 h, and IPTG was added to the wells using a liquid handling robot (Dispendix iDot). The volume of each well was normalized to 125 μL with M9 media. The plate was then sealed with a breathable membrane (Breathe-Easy Z380059) and placed back into the plate reader for a further 16 h at 37 °C and shaking. Measurements of absorbance (600 and 700 nm) and fluorescence (excitation 488 nm and emission 530 nm) were taken every 20 min. The data were processed using FlopR[32] to normalize and convert to standard units. The processed data were plotted using custom scripts written in R[33] using packages: ggplot2[34].

### Agar plate field experiments

Five microlitres of overnight cultures were diluted in 5 mL of fresh M9 media with the relevant antibiotics. The cultures were then allowed to grow for 2 h at 37 °C and shaking at 200 rpm. 1× M9 agar was made by mixing equal volumes of warmed 2× M9 media and molten 3% agar and kept in a water bath at 50 °C to prevent it from solidifying. In total, 20 mL of the M9 agar was dispensed into a one-well plate (Greiner 670102) and allowed to dry for 30 min. A 10 mL layer of M9 agar, inoculated with the relevant strain, was then poured over the top. This layer consisted of 5 mL of 2× M9 media with antibiotics, 2.5 mL of 3% agar, and 2.5 mL of culture. This results in an agar density of 0.75% which makes it easier to produce a level layer before it solidifies and enables the bacterial lawn to grow more evenly. The plates were left to dry for a further 15 min. Channels were cut in the agar to split the plate into four equally sized, separated areas; roughly between columns 6 and 7, and between rows D and E of a 96-well plate layout. This was to prevent diffusion of inducer between each region of the plate, enabling four experiments to be run concurrently. One microlitre of droplets of 7.5 mM IPTG were dispensed at precise locations onto the surface of the bacterial lawns using a liquid handling robot (Opentrons

OT2). The plates were left to dry until the droplets were no longer visible; approximately 15 min. They were then placed into an incubator at 37 °C and imaged with a custom imager from loopbio at 0, 16, 18, and 20 h for growth (red light, intensity 0.7, exposure 4000) and fluorescence (blue light, intensity 3, exposure 20,000).

### Agar plate colony experiments

The wells of a 6-well plate (Greiner 657185) were filled with 3 mL of 1× M9 agar made as described above. The plate was allowed to dry for 20 min. The optical density of overnight cultures was measured at 600 nm, and the cultures were diluted to an OD600 of 0.3. For the characterization experiments, 1 μL of the diluted culture was dispensed, at the positions shown in Supplementary Fig. 5, onto the surface of the agar using a liquid handling robot (Opentrons OT2). For the logic experiments, dispensing positions were determined using the Macchiato algorithm described above. One microlitre of IPTG, at 7.5 mM for logic experiments and at various concentrations for characterization experiments, was dispensed in the same way. For the characterization experiments, the plates were left to dry until the droplets were no longer visible; approximately 15 min. For the logic experiments, some IPTG locations were on top of the culture locations. As such, we first dispensed the culture and allowed it to dry until the droplets were no longer visible. Then we dispensed the IPTG and again allowed the plate to dry until the droplets were no longer visible. They were then placed into an incubator at 37 °C and imaged with a custom imager from loopbio at 0, 16, 18, and 20 h for growth and fluorescence as described above.

### Reporting summary

Further information on research design is available in the Nature Portfolio Reporting Summary linked to this article.

## Data availability

All data and code used to produce figures are available from Zenodo (https://zenodo.org/records/11262071). Plasmids for the AHL sender and receivers are available from Addgene.

## Code availability

Code for Macchiato is available from GitHub.

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

## Acknowledgements

We would like to thank: Chunbo Lou (Shenzhen Institute of Advanced Technology) and Yolanda Schaerli (University of Lausanne) for providing plasmids and strains used in this work; European Research Council (ERC) under the European Union's Horizon 2020 research and innovation program grant 770835 (A.J.H.F., N.J.T., K.Y.W., L.D., Q.H.O., G.J., J.W.R., C.P.B.); Wellcome Trust 209409/Z/17/Z (C.P.B.); Biotechnology and Biological Sciences Research Council (BBSRC) through the London Interdisciplinary Doctoral Program (L.R.); Engineering and Physical Sciences Research Council (EPSRC) through the BioDesign Engineering Centre for Doctoral Training (K.J.Y.Z.).

## Author contributions

Study conception and design: A.J.H.F., N.J.T., K.Y.W., C.P.B. Funding and supervision: A.Z., C.P.B. Paper writing: A.J.H.F., N.J.T., K.Y.W., C.P.B. Paper editing: A.J.H.F., C.P.B. Theoretical results: A.J.H.F., N.J.T., E.L. Mathematical modeling: A.J.H.F., N.J.T., L.R. Experimental work: A.J.H.F., K.Y.W., L.D., Q.H.O., G.J., J.W.R., K.J.Y.Z., L.R.

## Competing interests

The authors declare no competing interests.
