## [Peer Review File · Nature Communications]

Reviewers' Comments:

Reviewer #1:

Remarks to the Author:

This study presents a novel molecular computational framework for constructing logic functions. While the underlying concept is interesting, the functions implemented in this study have been repetitively presented and are limited to three inputs with widely known logic functions (majority, AND, OR, etc.).

Major comments:

- The authors should provide a straightforward explanation for their computational concept, such as clarifying why the system computes an AND logic gate when the inputs are at a distance from the cells, as opposed to computing an OR logic gate when they are in proximity to the cells.
- My main concern about this concept is that the locations of the inputs and cells are contingent on the logic functions. The inputs are defined by their location, time, and magnitude. The authors can not change both the location of cells and inputs. Cells should be fixed and inputs can be changed.
- Secondly, although the authors' approach to constructing computing functions is innovative, the specific logic functions they employed are not novel, and they have not demonstrated a new function that hasn't already been covered in prior studies. I believe it would be valuable for the authors to integrate at least one original function into their work
- Third -scalability- how this concept can scale-up, including more inputs and layers
- Fourth- how this concept can be used in real applications – since agar is not used their.
- In the introduction missing important manuscripts presenting new molecular computing diagram, especially, page 2 from line 17 till

Minor

1. Page 2 line 28, "In contrast to previous work", should indicate which work
2. The authors can not claim that "This approach also guarantees that any function can be implemented using only one communication molecule, solving the wiring problem", only if authors show scalability (number of inputs and computational complexity). In this case, I think simulation results will be enough.
3. I recommend that the authors provide an explanation in the main narrative regarding the implementation of their bandpass filter
4. The authors wrote "One might, therefore, expect the diameter of the ring formed by the bandpass cells to change with concentration and time. Indeed, with higher inducer droplet concentrations, the ring size increases. However, the observable position of the ring is temporally static, indicating that at some point – perhaps upon reaching a particular growth phase – the cellular response is established and fixed – is this feature related to any inducer to any media ? How the authors can explain this phenomena
5. The fold changes in the AND logic gates is very low (less than 3) and other gates. I think it will be very difficult to scale the system with such values in the future.
6. Two of these – NAND and IMPLY – require two output colonies, which are computed from combinations of NOT and IDENTITY functions using the OR paradigm described previously . Figure 5, I do think that the authors can merge the signals from two colonies and claim that they have a logic function. In their concept location is a very important parameter.
7. The biosensing application is kind of interesting, but can the authors explain why they decide to detect Arabinose and Lactate?, is there any clinical application for these two inputs?
8. Throughout the manuscript, the authors consistently emphasized a significant advantage of their concept, namely, its reduced need for numerous wires, requiring only one. This holds true as long as the system is scalable. However, if their concept lacks scalability, I find it challenging to identify a substantial advantage.
9. I recommend that the authors illustrate the time-dependent kinetics response of their system, focusing on a specific circuit, preferably a circuit (XOR) with the bandpass, and demonstrate its dynamics at various time points.
10. How the sharpness and the threshold of the activation functions affect their circuit response?

Reviewer #2:

Remarks to the Author:

In this study, the authors present a methodology for the implementation of multicellular logic circuits using a minimal library of genetically modified bacteria and utilizing space as a computational element. In this implementation, they use a diffusible molecule called "morphogen" to perform various computations, and different response functions to this morphogen by the receiver cells.

While this study technically sounds, it does not appear to provide significant new advancements in the field of logical digital circuit creation, from a conceptual standpoint, compared to the current state of the art. Consequently, it may be considered a technical work exploring an alternative implementation to those previously published.

Based on the presented results, it does not seem evident that this alternative implementation offers significant advantages over previously published ones, either in reducing the size of the cellular library or in the complexity of the different logical functions they implement. I will now present the arguments on which I base this opinion.

1. In this work, a morphogen molecule is used as a key element for computation. This solution has already been explored in Mogas et al. (Nat. Comms <https://doi.org/10.1038/s41467-021-21967-x>).

2. Unlike Mogas et al., instead of using modulating elements to modify morphogen diffusion gradients in response to external inputs, here, the morphogen is applied at different positions in space. The distance between the application position of the morphogen and the receiver cell determines the concentration that this cell will detect. This approach has already been explored in Tamsir et al. (doi: 10.1038/nature09565) and in Mogas et al. Surprisingly, this part of the paper omits the use of cells capable of transforming external inputs into the morphogen. Consequently, what distinguishes a logical 0 from a 1 depends solely on the position at which the morphogen is applied.

3. What is used are modulating elements for the response of receiver cells to the morphogen. Specifically, it is demonstrated that all computations can be implemented using bandpass or bandstop filters. While this is different from the implementation in Mogas et al., which used Identity and NOT functions as modulating elements, it should be noted that these two functions are actually highpass and lowpass filters, respectively, given the transfer function profiles that characterize the response of these modulating cells. Likewise, it seems clear that both a bandpass and a bandstop circuit can be obtained by combining a highpass and lowpass filter (in the case of bandpass, this is demonstrated in Mogas et al.). Since the ID and NOT functions, combined with spatial segregation, define a functionally complete set of Boolean algebra (as demonstrated in Macia et al. (<https://doi.org/10.1371/journal.pcbi.1004685>)), logically, the combination of highpass and lowpass also defines a functionally complete set, and any Boolean function can be implemented with them. However, from a computational complexity standpoint, bandpass and bandstop circuits are more complex than highpass and lowpass ones.

In addition to these arguments that, in my opinion, question whether this work represents a significant advancement in the field conceptually, it also does not appear that this implementation provides a solution to the current limitations in scaling the computational complexity of cellular circuits. The presented implementations are limited to basic logic circuits with up to 3 inputs, which are currently surpassed.

I dare to suggest that this implementation could be further developed if the authors were to address the implementation of analog circuits using combinations of different morphogen diffusion

gradients.

In conclusion, I believe that this work, being more of a technical contribution than a conceptual one, should be published in a different type of journal, perhaps one specialized in the field of cellular computing.

Reviewer #1 (Remarks to the Author):

This study presents a novel molecular computational framework for constructing logic functions. While the underlying concept is interesting, the functions implemented in this study have been repetitively presented and are limited to three inputs with widely known logic functions (majority, AND, OR, etc.).

Major comments:

- The authors should provide a straightforward explanation for their computational concept, such as clarifying why the system computes an AND logic gate when the inputs are at a distance from the cells, as opposed to computing an OR logic gate when they are in proximity to the cells.

We have added to the following text to the manuscript and updated figure 1 in order to clarify this:

"All of our inputs produce the same diffusible molecule, so where there is overlap in the diffusion field, the resulting concentration is additive. These inputs produce a unique diffusion field for each of the four input states."

"For a simple receiver, in which the cells turn on when the concentration is above a threshold, when the two input sources are close to each other, within the overlapping region the concentration will be high enough to trigger the output when one or both inputs are present (Fig. 1E). If the two input sources are moved further apart, at the center of the overlapping region the concentration will not be high enough to trigger the output when only one source is present but when both sources are present, the sum of the concentration will push the concentration above the output threshold. As such, we are able to change the logic function computed, by moving the sources of the diffusible molecules."

- My main concern about this concept is that the locations of the inputs and cells are contingent on the logic functions. The inputs are defined by their location, time, and magnitude. The authors can not change both the location of cells and inputs. Cells should be fixed and inputs can be changed.

We thank the reviewer for this comment but we are unsure what is meant. In our system, the logic function is encoded in the relative positions of the inputs to the outputs. Therefore, it is intrinsic to our system that both the inputs and outputs are movable. If the concern is that we should demonstrate changing the logic function by holding the output colony in position and changing the position of the inputs ("Cells should be fixed and inputs can be changed."), this is exactly what we did in Figure 2.

- Secondly, although the authors' approach to constructing computing functions is

innovative, the specific logic functions they employed are not novel, and they have not demonstrated a new function that hasn't already been covered in prior studies. I believe it would be valuable for the authors to integrate at least one original function into their work

We started by demonstrating all of the 2-input gates that were achievable with the activation functions that we had. We agree that this is repetitive of prior work but it demonstrates the foundation. The 3-input gates that we produced were chosen so that we could compare the complexity of our system to one that would use gene regulatory networks (i.e. Cello). The specific gates spanned the Cello complexity in terms of number of parts required. The sum=1 function wasn't performed in Cello and the 2-to-1 MUX that we implemented was IF C THEN B ELSE A rather than Cello's IF B THEN C ELSE A. In addition, the two receiver gates in figure 4 are composed of gates which weren't demonstrated in Cello (0x0F, 0x0A, 0x48, and 0x66).

However, we have now added two additional gates, in an updated figure 5, that haven't previously been shown. We have added an extra 3-input logic gate that encodes the rule 110 cellular automata logic – a Turing complete rule set. We have also added a 4-input logic gate, which, to our knowledge, has not been previously demonstrated.

- Third -scalability- how this concept can scale-up, including more inputs and layers

We thank the reviewer for their comment. The question of scaling up inputs is an important one that we did touch on in the discussion. However, to more completely answer this comment, we have added an implementation of a 4-input logic function to demonstrate adding more inputs and we have updated the discussion with the following text.

"Our theory shows that we need, at most, $n-1$ output colonies to produce all n -input logic functions. This scales favourably compared to other biological computation approaches. However, as we add more inputs and outputs, it is feasible that it is not possible to find a valid pattern in 2-dimensions, as moving one input relative to one output also moves it relative to all other outputs. This may be alleviated by the redundancy present in the four activation functions i.e. it is possible to produce an OR gate in multiple ways using highpass or bandpass activation. Further, moving to three-dimensional devices would relieve this limitation, and provide an avenue for the development of computational bio-materials (25, 26). An additional challenge when we scale the number of inputs is that we amplify our weakness at computing AND logic. For two inputs, our receivers need to flip from the OFF to ON state with a doubling of signal concentration e.g. the difference between having one input present (01) and both inputs present (11). However, for three inputs the same behaviour needs to occur with only a 50% increase in signal concentration (011 to 111) and for four inputs it is only a 25% increase (0111, 1111). In order to ensure that we can achieve this, the activation

functions of our receiver colonies need to be as step like as possible. Approaches such as toggle switch topologies and recombinases have been demonstrated in the past to improve digital response.”

Regarding the scaling of layers, this is theoretically possible by having intermediate receivers that produce diffusible signals similarly to our biosensor inputs in figure 5. However, the logic decomposition approach that we adopted, based on the disjunctive normal form, specifically enables us to produce complex logic with only a single layer. This is described in detail in the section “Two-level logic enables the construction of arbitrary digital functions”.

- Fourth– how this concept can be used in real applications – since agar is not used their.

We thank the reviewer for this comment. While we agree that it is important to consider how a new system may be applied in future, this paper’s focus was to demonstrate a new mode of spatial biocomputing. Therefore we believe it is beyond the scope of the current work.

- In the introduction missing important manuscripts presenting new molecular computing diagram, especially, page 2 from line 17 till

We believe this comment has been cut short. If the reviewer has particular references in mind, we would be glad to include them. We believe we have included the two key spatial biocomputing references in Macia et al. 2016 Plos Comp Biol, and Mogas-Diez et al. 2021 Nat Commun.

Minor

1. Page 2 line 28, “In contrast to previous work”, should indicate which work

We were referring to the work cited in the previous paragraph. We have now included those references in line.

2. The authors can not claim that “This approach also guarantees that any function can be implemented using only one communication molecule, solving the wiring problem”, only if authors show scalability (number of inputs and computational complexity). In this case, I think simulation results will be enough.

We have added an in vitro implementation of a 4-input logic gate and lay out mathematically in the supplementary how our approach scales with the number of inputs.

3. I recommend that the authors provide an explanation in the main narrative regarding the implementation of their bandpass filter

The IPTG bandpass receiver was implemented in previous work by Chunbo Lou's group Zong et al. 2017 Nat Commun. As it was not created by us, we do not feel that we should discuss its implementation in detail. We included our characterisation of its behaviour in the supplementary. To provide more explanation as to how the bandpass functions, we have added the following in the main text.

"The bandpass behaviour is produced by a T7 promoter that is repressed by PhIF transcription factor (Fig. S3). T7 RNAP and PhIF are expressed from IPTG inducible promoters in the genome and on a plasmid respectively. In the absence of IPTG, no T7 RNAP is expressed and the output promoter is not activated. At intermediate levels of IPTG, T7 RNAP is produced and transcription from the output promoter occurs. At high IPTG levels, PhIF is expressed at high enough concentration to inhibit transcription from the output promoter. The highpass receiver is the same system but without the PhIF expressing plasmid."

4. The authors wrote "One might, therefore, expect the diameter of the ring formed by the bandpass cells to change with concentration and time. Indeed, with higher inducer droplet concentrations, the ring size increases. However, the observable position of the ring is temporally static, indicating that at some point – perhaps upon reaching a particular growth phase – the cellular response is established and fixed – is this feature related to any inducer to any media? How the authors can explain this phenomena

We believe that this is due to phenotypic changes in the receiver cells as they exit exponential growth. It appears from both liquid culture and solid culture experiments that the cells response is fixed by the concentration of inducer that the cells are exposed to during exponential phase. While we believe it would be interesting to explore this behaviour further, it is beyond the scope of this work.

5. The fold changes in the AND logic gates is very low (less than 3) and other gates. I think it will be very difficult to scale the system with such values in the future.

Please see response to question on scalability above.

6. Two of these – NAND and IMPLY – require two output colonies, which are computed from combinations of NOT and IDENTITY functions using the OR paradigm described previously. Figure 5, I do think that the authors can merge the signals from two colonies and claim that they have a logic function. In their concept location is a very important parameter.

We are not sure what the reviewer means here. In our paradigm, because the output is GFP signal, if any of the outputs are ON, we regard the logic function to be ON. This is detailed in the section "Two-level logic enables the construction of arbitrary digital functions"

7. The biosensing application is kind of interesting, but can the authors explain why they decide to detect Arabinose and Lactate?, it there any clinical application for these two inputs?

Lactate, or lactic acid, is a metabolite produced during anaerobic metabolism and is found in elevated levels in tumours. Arabinose is a standard inducer used in engineering biology research. We used these to show that both standard inducers and important metabolites could be integrated into the system.

8. Throughout the manuscript, the authors consistently emphasized a significant advantage of their concept, namely, its reduced need for numerous wires, requiring only one. This holds true as long as the system is scalable. However, if their concept lacks scalability, I find it challenging to identify a substantial advantage.

Please see our response to the question of scalability above.

9. I recommend that the authors illustrate the time-dependent kinetics response of their system, focusing on a specific circuit, preferably a circuit (XOR) with the bandpass, and demonstrate its dynamics at various time points.

We thank the reviewer for this suggestion. We have added the time-dependent kinetics of single bandpass colonies to Supplementary Figure 5 and added Supplementary Figure 13 showing the behaviour of the XOR gate at different timepoints.

10. How the sharpness and the threshold of the activation functions affect their circuit response?

Please see our response to the question of scalability above.

Reviewer #2 (Remarks to the Author):

In this study, the authors present a methodology for the implementation of multicellular logic circuits using a minimal library of genetically modified bacteria and utilizing space as a computational element. In this implementation, they use a diffusible molecule called "morphogen" to perform various computations, and different response functions to this morphogen by the receiver cells.

While this study technically sounds, it does not appear to provide significant new advancements in the field of logical digital circuit creation, from a conceptual standpoint, compared to the current state of the art. Consequently, it may be considered a technical work exploring an alternative implementation to those previously published.

Based on the presented results, it does not seem evident that this alternative implementation offers significant advantages over previously published ones, either in reducing the size of the cellular library or in the complexity of the different logical functions they implement. I will now present the arguments on which I base this opinion.

1. In this work, a morphogen molecule is used as a key element for computation. This solution has already been explored in Mogas et al. (Nat. Comms <https://doi.org/10.1038/s41467-021-21967-x>).

We agree with the reviewer that diffusion has been used before for logic computation. We already included references to Mogas et al. and Macia et al. in the manuscript.

2. Unlike Mogas et al., instead of using modulating elements to modify morphogen diffusion gradients in response to external inputs, here, the morphogen is applied at different positions in space. The distance between the application position of the morphogen and the receiver cell determines the concentration that this cell will detect. This approach has already been explored in Tamsir et al. (doi: 10.1038/nature09565) and in Mogas et al.

We disagree with the reviewer that distance has been used in such a way before. In Tamsir et al., the distance is not varied – all colonies are spaced 7mm apart. Instead, they use multiple different AHL molecules as wires between their computational colonies. This means that to construct the sixteen 2-input logic gates, they have to engineer 8 different cell types. This is also not simply expandable to 3-input logic.

As stated by the reviewer, Mogas et al. use morphogen modifying cells as the main means to modify the morphogen concentration seen by the output colony. Distance is not explicitly used as a parameter for changing logic functions, although distances were altered to accommodate the strengths of different senders. Indeed, distance seems to

provide a challenge when multiple morphogen modifying cells are used, resulting in the need to introduce an amplifying population.

Surprisingly, this part of the paper omits the use of cells capable of transforming external inputs into the morphogen.

We omit the morphogen generating cells from the early parts of the paper as these are not required for demonstrating this paradigm of digital logic computation. Because computations are encoded solely by the spatial pattern of inputs and outputs, and the activation function of the output, it does not matter that an input is a droplet of inducer or a sender colony.

Consequently, what distinguishes a logical 0 from a 1 depends solely on the position at which the morphogen is applied.

It is true that the computation depends solely on the diffusion field, that is the position of the sources of the morphogen relative to the receiver colonies. For the majority of the paper, the source of the function input is also the source of the morphogen i.e. droplets of IPTG. However, for the final part of the paper, figure 6, this is decoupled so that the inputs are globally applied and the morphogen sources are bio-sensing sender cells.

3. What is used are modulating elements for the response of receiver cells to the morphogen. Specifically, it is demonstrated that all computations can be implemented using bandpass or bandstop filters. While this is different from the implementation in Mogas et al., which used Identity and NOT functions as modulating elements, it should be noted that these two functions are actually highpass and lowpass filters, respectively, given the transfer function profiles that characterize the response of these modulating cells. Likewise, it seems clear that both a bandpass and a bandstop circuit can be obtained by combining a highpass and lowpass filter (in the case of bandpass, this is demonstrated in Mogas et al.). Since the ID and NOT functions, combined with spatial segregation, define a functionally complete set of Boolean algebra (as demonstrated in Macia et al. (<https://doi.org/10.1371/journal.pcbi.1004685>)), logically, the combination of highpass and lowpass also defines a functionally complete set, and any Boolean function can be implemented with them. However, from a computational complexity standpoint, bandpass and bandstop circuits are more complex than highpass and lowpass ones.

We agree with the reviewer that bandpass and bandstop receivers, are indeed more complex to genetically engineer. However, we justify their use in two ways.

The use of the bandpass and bandstop allow us to perform all logic functions with a single layer. This is not possible with just the highpass and lowpass (ID and NOT) as stated in the manuscript – and described in detail in the supplementary material.

Additionally, they only ever need to be constructed once as they only ever need to respond to a single molecule. This is one of the difference and key advantages of our

system compared to Mogas et al., Tamsir et al., and other approaches to date – given only four engineered cell types, we can compute arbitrary logic functions. In fact we only require two - the bandpass and bandstop - but including the highpass and lowpass provides more flexibility in spatial positioning. Every system that has come before requires segregation of components through the use of liquid chambers (Macia et al.), confined strips of paper (Mogas et al.), or multiple communication molecules (Tamsir et al.).

In addition to these arguments that, in my opinion, question whether this work represents a significant advancement in the field conceptually, it also does not appear that this implementation provides a solution to the current limitations in scaling the computational complexity of cellular circuits. The presented implementations are limited to basic logic circuits with up to 3 inputs, which are currently surpassed.

We thank the reviewer for their comment. We have added an in vitro implementation of a 4-input logic function. While we do believe that we are confronted with some scaling problems with this approach, they are different scaling problems to existing approaches. We have demonstrated theoretically that our approach scales favourably compared to that of Mogas-Diez et al – for an n-input logic function we require at most n-1 receivers versus $2^{(n-1)}$ branches. There is also no further requirement for engineering new strains beyond the four required activation functions. Where we may face scaling issues is in our ability to find patterns that allow all input and output positions to be correctly distanced from each other. A further issue with scaling is our ability to perform AND like behaviour as the number of inputs increases. We have updated the discussion regarding scaling with the following text:

“Our theory shows that we need, at most, n-1 output colonies to produce all n-input logic functions. This scales favourably compared to other biological computation approaches. However, as we add more inputs and outputs, it is feasible that it is not possible to find a valid pattern in 2-dimensions, as moving one input relative to one output also moves it relative to all other outputs. This may be alleviated by the redundancy present in the four activation functions i.e. it is possible to produce an OR gate in multiple ways using highpass or bandpass activation. Further, moving to three-dimensional devices would relieve this limitation, and provide an avenue for the development of computational bio-materials (25, 26). An additional challenge when we scale the number of inputs is that we amplify our weakness at computing AND logic. For two inputs, our receivers need to flip from the OFF to ON state with a doubling of signal concentration e.g. the difference between having one input present (01) and both inputs present (11). However, for three inputs the same behaviour needs to occur with only a 50% increase in signal concentration (011 to 111) and for four inputs it is only a 25% increase (0111, 1111). In order to ensure that we can achieve this, the activation functions of our receiver colonies need to be as step like as possible. Approaches such as

toggle switch topologies and recombinases have been demonstrated in the past to improve digital response.”

I dare to suggest that this implementation could be further developed if the authors were to address the implementation of analog circuits using combinations of different morphogen diffusion gradients.

We thank the reviewer for their comment. This is indeed a development that we are pursuing but it is beyond the scope of this paper.

In conclusion, I believe that this work, being more of a technical contribution than a conceptual one, should be published in a different type of journal, perhaps one specialized in the field of cellular computing.

Reviewers' Comments:

Reviewer #1:

Remarks to the Author:

Although the authors have addressed many of my comments, there are still some that remain unanswered

1. The primary challenge in scaling and utilizing their systems is the low ON/OFF ratio observed in most of their circuits. Some of their systems demonstrate a ratio of only two, which could pose significant issues for computation when factoring in noise. It will be hard to scale systems with very low ON/OFF ratio.

2. The majority circuit (Fig. 4c) did not exhibit truly digital behaviour – it is more analog behavior

3. The AND logic gate (Figure 2B) does not appear to be functioning correctly. Even upon visual inspection of the images (10) and (01), a shadow that is similar to 11 can be observed, indicating an analog behaviour rather than the expected digital output.

4. Please check the image in Fig. 4B which shows very strong bands but the ratio ON/OFF is very low

5. The authors acknowledged that their systems lack real-world applications, stating that it is beyond the scope of their study. Therefore, please remove the “biosensor” from the manuscript, abstract, and Figure 6

6. How robust is their design concerning colony location?. For instance, in the OR logic gate, even slight changes in the colony locations may lead to significant fluctuations in the outputs.

Conversely, the AND logic gate should ideally exhibit greater robustness. Please confirm this experimentally.

7. Regarding Figure 5, I disagree with the authors' response and their claim that they can extract GFP from colonies placed in different locations and treat them as a single input. If they can explain how in reality they can do such an operation

Overall, while the concept of utilizing summation and thresholding has been previously explored, the idea presented in this study – using space for computation- is interesting

Reviewer #2:

Remarks to the Author:

Following the improvements made in this study, I believe it is suitable for publication.

Reviewer #1 (Remarks to the Author):

Although the authors have addressed many of my comments, there are still some that remain unanswered

We thank the reviewer for their comments and their continued efforts to improve our manuscript. We appreciate the acknowledgement that using space for computation is novel.

Our manuscript introduces this new approach for biocomputing based around the spatial computing concept. We have developed the theoretical foundations for the architecture and constructed a proof-of-principle experimental implementation with an associated mathematical model. We believe that remaining comments are focussed on optimisation of aspects of the *implementation* rather than fundamental flaws in the *architecture*. Points 1-4 reflect a current weakness in the implementation of our highpass receiver, specifically the steepness of its activation making AND behaviour difficult. We have been open about these limitations in the manuscript and suggested paths to improvement in the discussion. As is always the case with proof-of-principle work, there is much to do in the future to improve upon it.

1. The primary challenge in scaling and utilizing their systems is the low ON/OFF ratio observed in most of their circuits. Some of their systems demonstrate a ratio of only two, which could pose significant issues for computation when factoring in noise. It will be hard to scale systems with very low ON/OFF ratio.

The weakest circuit we present is the Majority circuit with a signal-to-noise ratio of 2.29. This is of course lower than desired. It should be noted however, that this is already subject to both biological and experimental noise, and that this digital function is very complex if one were to reproduce it as a genetic circuit (49 genetic parts in Cello).

2. The majority circuit (Fig. 4c) did not exhibit truly digital behaviour – it is more analog behavior

It is true for the Majority, and some other circuits, that some of the ON responses are lower than other ON responses. This is predicted by our model and intuitive given the activation function of our highpass receiver – 3-inputs produces a higher response than 2-inputs. Our reported signal-to-noise ratios always use the lowest ON state and the highest OFF state, giving the most conservative score.

3. The AND logic gate (Figure 2B) does not appear to be functioning correctly. Even upon visual inspection of the images (10) and (01), a shadow that is similar to 11 can be observed, indicating an analog behaviour rather than the expected digital output.

It is true that the intermediate states (in which one of either input is present) shows a faint ON state in the colony. We deliberately chose the exposure on the camera in order to highlight the previously discussed weakness of our current highpass implementation.

4. Please check the image in Fig. 4B which shows very strong bands but the ratio ON/OFF is very low

This is the same effect as the examples above. The AND response (A AND C) is weaker than the OR response with input B. We could have moved the position of input B further away from the receiver to reduce the strong B OR X responses so they are closer to the A AND C response, but we chose to maximise the output.

5. The authors acknowledged that their systems lack real-world applications, stating that it is beyond the scope of their study. Therefore, please remove the “biosensor” from the manuscript, abstract,

and Figure 6

We take the reviewer's point around real-world application. We have changed the wording in the main text and abstract to remove references to this being a biosensor/diagnostic device.

6. How robust is their design concerning colony location?. For instance, in the OR logic gate, even slight changes in the colony locations may lead to significant fluctuations in the outputs. Conversely, the AND logic gate should ideally exhibit greater robustness. Please confirm this experimentally. Overall, while the concept of utilizing summation and thresholding has been previously explored, the idea presented in this study – using space for computation- is interesting

The lawn assays in Figure 1 and the supplementary information Figure 4 demonstrate the sizes of the regions in which we can place a receiver and produce a given function. The OR logic is remarkably robust for our highpass implementation; any arrangement in which the receiver is within 10mm of both inputs will produce an OR gate with >5 SNR (4 discrete colony distances = 20 discrete input positions around the receiver, for our 384-position layout). However, for the AND gate there is only one distance that we can use with our 384-position layout that is able to give us a satisfactory SNR. The image below shows the positions in which inputs can be placed relative to a single output in order to achieve an OR gate (orange) or AND gate (grey).

7. Regarding Figure 5, I disagree with the authors' response and their claim that they can extract GFP from colonies placed in different locations and treat them as a single input. If they can explain how in reality they can do such an operation

Our computational device currently consists of inputs and output colonies placed within a single well of a six-well plate. To determine the output state of the system we take an image of the entire well, not of each individual colony. Therefore, we simply need to observe whether there is any fluorescence being expressed on the device (across a well) to determine if the overall state is ON. For example, the image below shows the output colonies for five different input states of the rule 110 logic gate shown in Figure 5. It is clear that there are inputs on in wells A1-3 and B2, but not in B1.

Rule 110

A1: 001

A2: 010

A3: 011

B1: 100

B2: 101

B3: control